# A Sacrificial Route for Soft Porous Polymers Synthesized via Frontal Photo-Polymerization

**DOI:** 10.3390/polym12051008

**Published:** 2020-04-27

**Authors:** Alexandre Turani-i-Belloto, Thomas Brunet, Alexandre Khaldi, Jacques Leng

**Affiliations:** 1Laboratory of the Future (CNRS/SOLVAY), University of Bordeaux, 178 avenue du Docteur Schweitzer, 33600 Pessac, France; alex.turani@outlook.fr (A.T.-i.-B.); alexandre.khaldi@imt-atlantique.fr (A.K.); 2Institut d’Ingénierie et de Mécanique, (CNRS/Bordeaux INP), University of Bordeaux, 351 Cours de la Libération, 33400 Talence, France; thomas.brunet@u-bordeaux.fr; 3IMT Atlantique Bretagne-Pays de la Loire, Technopôle Brest-Iroise, 29200 Brest, France

**Keywords:** hydrogels, photo-polymerization, sacrificial particles, porous polymers, microfluidics

## Abstract

Within the very large range of porous polymers and a related immense scope of applications, we investigate here a specific route to design *soft porous polymers with controlled porosity*: we use aqueous-based formulations of oligomers with mineral particles which are solidified into a hydrogel upon photo-polymerization; the embedded particles are then chemically etched and the hydrogel is dried to end up with a soft porous polymeric scaffold with micron-scale porosity. Morphological and physical features of the porous polymers are measured and we demonstrate that the porosity of the final material is primarily determined by the amount of initially dispersed sacrificial particles. In addition, the liquid formulations we use to start with are convenient for a variety of material forming techniques such as microfluidics, embossing, etc., which lead to many different morphologies (monoliths, spherical particles, patterned substrates) based on the same initial material.

## 1. Introduction

The porosity in polymers may emerge from a variety of physical or chemical routes, e.g., phase separation [1], foaming [2], emulsion templating [3], sacrificial particles [4]… It leads to important functionalities such as light weight, insulation, permeability, etc. that find numerous applications. Specific material forming processes may also add up to more functionalities: spherical particles are useful for instance for chromatography, drug delivery, acoustics [5], whereas thin films of can be used as membranes for filtration; fibers are useful for fabric and promising for tissue engineering [6]; eventually, porous polymers become increasingly appealing for 3D printing [7]. In most of these applications, the features of the porosity are of prime importance, namely and at minima its size, distribution, connectivity, specific surface, etc.

Here, we follow a soft route to create porous polymers materials with controllable porosity whereby sacrificial solid particles are removed after the photo-curing of a hydrogel. We study the critical steps that lead to a solid porous polymer starting from an aqueous dispersion: optimization of the formulation containing the oligomers, the sacrificial particles and water; photo-curing of turbid dispersions; removal of the particles [4]; drying (not shown here); and features of the final porosity. As a main result, we provide evidence that the porosity of the dried polymer perfectly relates to the initial quantity of the sacrificial particles; additionally, connected or disconnected porosity could possibly be produced depending on the initial formulation. Eventually, due to the specific rheological nature of the formulations (low viscosity, negligible yield-stress), we demonstrate that these fluids can be formed under a large variety of structures using advanced techniques such as for instance soft embossing and microfluidics.

## 2. Materials and Methods

### 2.1. Chemicals

The core formulation contains an oligomer, poly(ethylene glycol)-diacrylate (PEG-DA, $M_{n}=700$ g·mol^−1^, density $\rho =1.12\times 10^{3}$ kg·m^−3^, Merck, Darmstadt, Germany), a photo-initiator (HMP, 2-hydroxy-2-methylpropiophenone, $M_{w}=164.02$ g·mol^−1^, density $\rho =1.077\times 10^{3}$ kg·m^−3^, Merck, Darmstadt, Germany, also known as Darocur 1173) and ultra-pure water.

To this formulation, we add two ingredients: (1) Bare micronized calcite (CaCO_3_) particles that will act as a solid sacrificial porogen: Microcarb 60, kindly provided by Reverté Productos Minerales, Spain, volume-averaged diameter ≈1.3 μm, distributed between 0.2 and 10 μm, molar mass $M_{w}=100.09$ g·mol^−1^, density $\rho =2.71\times 10^{3}$ kg·m^−3^; (2) A stabilizer: we studied two different polymeric stabilizers for the calcite: poly(sodium 4-styrenesulfonate) (PSS, $M_{w}=70\times 10^{3}$ g·mol^−1^, density $\rho =0.8\times 10^{3}$ kg·m^−3^, Merck, Darmstadt, Germany) known to disperse calcite particles during calcium salt precipitation [8,9] or poly(vinyl alcohol) (PVA, $M_{w}\approx 30\times 10^{3}$ g·mol^−1^, fully hydrolyzed, density $\rho =1.3\times 10^{3}$ kg·mol^−1^, Merck, Darmstadt, Germany) known as a highly hydrophilic compound due to hydroxyl groups yet with somewhat amphiphilic properties [10].

In practice, we prepare several stock solutions (a polymeric batch of PEG-DA/HMP and aqueous batches of PSS or PVA solutions); we add calcite to the aqueous batches in specific quantities, and finally add the polymeric batch in order to match the mass ratio we seek and to vary the volume fraction of calcite particles. We ensure that the final formulation contains PEG-DA/HMP/water in mass proportions 1/0.1/1.1 and the concentration of calcite particles will be given as a volume fraction ϕ in the final formulation. Thorough mechanical mixing followed by ultra-sound fragmentation is used before any post-treatment such as photo-polymerization.

After photo-polymerization of the formulation, calcite is removed through acidic treatment [4] using hydrochloric acid (HCl solution, 1 mol·L^−1^, Fisher Scientific, United States).

Silane are used to enhance adhesion of the formulation on glass slide (3-(trimethoxysilyl)propyl acrylate from Merck, Germany) or to reduce it (Aquapel, a commercial fluoroalkylsilane purchased online).

The poly(dimethylsiloxane) (PDMS) used for microfluidics is the Sylgard 184 Silicone Elastomer Kit.

The oil used for emulsification is the Fluorinert FC-40 purchased from Merck, Germany, and microfluidic and batch emulsions are stabilized with Pico-Surf purchased from Dolomite, United Kingdom. The microfluidic chips are made fluorophilic with Aquapel after plasma treatment and thorough oxygen degassing and argon regassing.

### 2.2. Photo-Polymerization

Flat and shallow monoliths are fabricated using frontal photo-polymerization (FPP) [11] performed as follows: about 2 mL of the formulation is pipetted and squeezed in between two glass slides (thickness 1 mm, fluorophilic) spaced with a 1 mm-thick spacer. The sample is then exposed to ultra-violet (UV) radiation using a collimated LED working at 365 nm (a UV-KUB1 from Kloé, France, delivering a tunable power density up to I≈500 W·m^−2^). The photo-polymerization yields a hydrogel monolith which turns from transparent to opaque as the calcite load is increased.

Concerning the systematic study of photo-polymerization, the main outcome is the thickness $z_{f}$ as a function of UV dose; the thickness is measured sidewise from a cross-section of a cut sample using stereo-microscopy or optical microscopy depending on the size of the sample.

Eventually, we also measured systematically the absorbance of all species in solution (except microparticles) and found that, at 365 nm, only the photo-initiator absorbs light and we measured the corresponding Beer–Lambert law as a function of concentration in water. From these measurements, we estimate that, in a typical formulation with no particle, the absorbance reads $A\left(l\right)=\mu_{PI}l$, where *l* is the optical path and $\mu_{PI}^{-1}\approx 2.5$ mm.

### 2.3. Microfluidics and Soft-Embossing

Advanced fabrication is achieved thought the use of templates such as microfluidic droplets (liquid templates) and moulds (solid templates) to engineer particles and patterned substrates respectively. Microfluidics and soft-embossing share a common technique, namely soft-photolithography based on PDMS material in order to fabricate microchannels and microtemplates.

## 3. Results

### 3.1. Optimized Formulation

The formulation was optimized according to the quality of the dispersion of calcite particles in the mixture of water and PEG-DA. Through the use of optical microscopy, static light scattering, and sedimentation velocity kinetics, we identified different sets of conditions for dispersing, or not, the particles in the formulation.

Figure 1 shows that PVA and PSS-based formulations behave quite differently. PVA leads to aggregated particles (which is barely different from native dispersion, with no coating), whereas PSS permits us to disperse quite perfectly the CaCO_3_ particles. Both results are interesting but, unless otherwise stated, we will focus here on PSS-based systems. The best PSS-based formulation actually contains ≈1 g·L^−1^ of PSS per mass % of calcite, which we set a rule whatever the actual concentration of calcite we introduce in the formulation; we believe it corresponds to some coating of the microparticles via electrostatic interactions [9] which is sufficient to provide a satisfactory dispersability.

Static light scattering confirms that, upon PSS addition in the formulation, the largest aggregates disappear (Figure 1), in which case we recover the size of particles given by the supplier.

We monitored the sedimentation kinetics in a Kynch-like experiment [12]—not shown here—to extract the sedimentation velocity at the level of the upper front along with the formation of a deposit at the bottom of the cuvette: the ‘best’ dispersion settles more slowly and also makes the most compact deposit. We acknowledge that these experiments are difficult to interpret due to polydispersity; however, the same range of PSS concentration results in correct dispersions of the calcite particles.

Eventually, we characterized the flow behavior of the PSS-based formulations, see Figure 2, and observe that at low concentration the formulation is essentially a Newtonian fluid and develops some yield stress above a volume fraction ϕ between 0.1 and 0.2. At 0.3 of calcite, the yield stress remains quite small (≈10 Pa) but increases significantly above (≈30 Pa at 0.4 volume fraction of calcite, Figure 2 insert), which makes the formulation more difficult to process with microfluidics for instance. An empirical extrapolation of the yield stress (line in Figure 2 insert) suggests that the critical yield stress develops here above ϕ≈0.15.

### 3.2. Preparation and Characterization of Porous Polymeric Monoliths

#### 3.2.1. Frontal Photo-Polymerization in the Presence of Calcite

We use photo-polymerization to create solids out of the aqueous-based formulations. Through the use of an UV light, the conversion of oligomers into cross-linked polymers stems from a reaction between photons and a photo-initiator: the photolysis of the latter under high energy photons produces radicals which then initiates acrylate polymerization that link oligomers of PEG-DA. In the unidimensional model geometry of frontal photo-polymerization (FPP) [11], the polymerized thickness is time-dependent and directly linked to the UV dose *E* (E=I×t, the product of UV power density *I* and time *t*), the quantity of oligomers, of photo-initiator, of inhibitors such as solubilized oxygen, UV light absorption due to the medium [13] and UV light scattering due to calcium carbonate particles [14].

The conversion of the formulation into a solid hydrogel is a key point in our work. Photo-polymerization is ubiquitous, especially in the field of microfluidic applications [15] as it is fast and simple, and we wondered how critical the presence of calcite particles was. We found in literature that photo-conversion of bare resins is described with a wealth of subtleties [11,13,16,17] unlike the case of loaded resins with minerals for instance, even though some significant pieces of work have been produced [14,18,19]. We believe the real mechanisms of FPP in strongly scattering media has not yet been fully appreciated.

The bare resin (with no calcite) was polymerized in a thick container with or without degassing (open symbols, Figure 3). We recover the so-called Jacob’s law:(1)$$z_{f}=\mu_{0}^{-1}\log\left(E/E_{c}\right)$$
where $z_{f}$ is the cured thickness (depth of polymerization), $\mu_{0}^{-1}$ is the absorption length related to the absorbers in the medium, E=I×t the UV dose to which the resin is exposed, and $E_{c}$ some critical dose at which the resin turns into a gel (the gel point $\varphi_{g}$ in Figure 3, left). In our case, we mentionned above that only the photo-initiators (PI) that trigger the chemistry absorb the UV radiation and therefore the absorption length is known for this specific chemistry: here, we find that $\mu_{0}^{-1}=\mu_{PI}^{-1}=2.5$ mm.

Degassing the sample does not alter the absorptivity of photo-initiators (the slope $z_{f}$ vs. *E* is constant) but reduces the presence of inhibitors such as dissolved O_2_ and therefore reduces the critical dose $E_{c}$ at which polymerization indeed starts.

For systems which are not degassed, adding mineral particles to the resin has two consequences on the curing depth kinetics: the cured depth *versus* the dose keeps a semi-logarithmic behavior, the slope of this pseudo Jacob’s law [$\mu {\left(\phi \right)}^{-1}$] drops according to the amount of calcite; more strikingly, the critical dose also drops as the amount of mineral increases: the latter result is already known in literature [20] and implies surprisingly that, the more turbid the formulation, the easier it is to initiate the polymerization. We developed a model that is able to describe roughly this behavior and which we will present elsewhere as a dedicated piece of work. However, and basically, the model is based on the sketch of Figure 3, left—picture inspired by Ref. [11]—and suggests that a proper way to describe the process must include the depth-dependent scattered intensity coupled with the photochemistry. We performed preliminary calculations based on Monte Carlo simulations which are encouraging (especially for the slope, see insert of Figure 3, right).

Master curves of Figure 3, right, are important to ensure the proper polymerization of formulated systems to be cured throughout their thickness, and ideally without gradient of conversion. Indeed, for studies on monoliths, we fabricated 1 mm-thick samples that may require a prohibitive dose to be cured in one shot when heavily loaded with calcite, see for instance ϕ=0.3 in Figure 3, right, which shows that reaching $z_{f}=0.5$ mm is tedious, especially because the kinetics is a logarithmic function of dose (thus $z_{f}∼\log t$). This systems shall then be cured for a (possibly very) long time and from the two sides to have a fully and homogeneously converted resin. We performed systematically this approach on our systems and this is also the reason which we restricted the thickness of the samples to 1 mm maximum.

#### 3.2.2. Characterization of Soft Porous Polymer

After FPP, we rinsed thoroughly the samples in pure water to remove traces of unreacted photo-initiators, dissolve the calcite in an acidic bath [4], rinse again the samples and dry them under atmospheric conditions (T≈20 °C, humidity ≈30–50%) or in an oven at 65 °C. Scanning Electron Microscopy (SEM) pictures provide evidence of non-connected porosity most likely due to the well dispersed calcite, see Figure 4b,c.

##### 3.2.2.1. Porosity Measured by Density Measurements

The density of a sample is related to its porosity: the measurement of the former yields the latter. Here, we reproduce Kovalenko’s experiments [21] and quote: *For porous samples, the air volume fraction was obtained using a home-made setup measuring the Archimedes force acting on samples immersed in a […] mixture. The obtained values were more precise than the ones obtained by direct calculation from the mass and the dimensions of the samples.* In a typical experiment, we measure initially the weight of a sample which is then forced to be immersed in a reference fluid, a fluorinated oil (perfluorinated polyether Fomblin–PFPE from Solvay) with a density of $1.91\times 10^{3}$ kg·m^−3^ and well-known non-wetting properties that prevent the oil to invade the sample. The introduction of the sample in the reference fluid modifies the mass of the reference fluid, read on a precision scale and converted into a density (See the web tutorial https://www.youtube.com/watch?v=uqnDGa1UN5o, last accessed 27 April 2020).

First, we measured the density of the liquid bare formulation (with no calcite) with an electronic densitometer and obtained $1.081\times 10^{3}$ kg·m^−3^, in between the ones of water and oligomer. Then, we obtained the density of the cured bare formulation: $\rho_{0}=\left(1.21\pm 0.02\right)\times 10^{3}$ kg·m^−3^ for the oven-dried PEG-DA polymer and $\rho_{0}=\left(1.18\pm 0.02\right)\times 10^{3}$ kg·m^−3^ for the air-dried one; these densities are higher than the one of the liquid formulation as the material shrinks upon polymerization and expels some water, i.e., undergoes some syneresis.

Then, we measured the density of four different porous monoliths (with an initial volume fraction of CaCO_3_ of $5\times 10^{-2}$, 0.1, 0.2, and 0.3). Figure 5, left, presents all the measured densities and the solid lines show that the densities can be understood in a first approximation on the basis of a linear combination of densities of the matrix $\rho_{0}$ and that of voids (where we neglect their contribution to the density, $\rho_{air}=1$ kg·m^−3^
$\ll \rho_{0}$ in the range of porosities of interest): $\rho \left(\phi \right)=\rho_{0}\left(1-\phi \right)$.

However, experiments show that, at a volume fraction below 0.1, densities are higher than expected, whereas, above 0.1, values are close to theoretical ones. In addition, the drying conditions do not seem to impact significantly the densities. A possible interpretation relies on the very hygroscopic nature of the polymer we use and which could well absorb atmospheric water even though it had been dried in an oven. We will see below that, when the system is degassed and observed under vacuum, we do not monitor any ‘lack’ of porosity.

##### 3.2.2.2. Porosity Measured by Image Analysis

Dried monoliths are cut and observed sidewise along the section using a scanning electron microscope (SEM tabletop TM3030Plus from Hitachi, Tokyo, Japan) after metal plating. During this stage, a 2mBar vacuum is exerted during about 5 min before metal deposition. Then, the samples are set in an SEM platform and left under vacuum during the observation (10 to 60 min). Figure 4a and b present monoliths and SEM pictures of monoliths with volume fractions of CaCO_3_ in the range 0–0.3: the porosity develops and increases with the amount of calcite. The measured porosity $\phi_{M}$ is obtained via image analysis using a Matlab program (R2016b, MathWorks, Natick, MA, USA) which identifies by threshold and segmentation the dark zones as being the pores and exclude interconnected pores; we thus obtain a statistical distribution of data: surface, equivalent diameter $d_{i}$, number of pores per unit of surface $N_{i}$ for each formulation. Eventually, we calculate the volume-averaged diameter $d_{v}=\left(\sum_{i=1}^{n}d_{i}^{4}N_{i}\right)/\left(\sum_{i=1}^{n}d_{i}^{3}N_{i}\right)$.

The measured porosity is then compared with the initial volume fraction of calcite ϕ, see black symbols in Figure 5 right, which shows an excellent correlation and which reflects that the drying of the samples is isometric and homogeneous, and preserves the aspect ratio of the sample. The microporosity is set by the amount of calcite initially dispersed in the formulation, at least up to ϕ=0.3. We also represent on this figure the porosity extracted out of the density: $\phi =1-\rho (\phi )/\rho_{0}$, and we obtain that, at fairly high volume fraction, all the measurements are in reasonable agreement.

##### 3.2.2.3. Porosity and Sound Speed

The propagation of elastic waves in a solid matrix is controlled by two material parameters, namely its density and some elastic modulus, and accepts longitudinal and transverse waves. The case of soft porous materials has been thoroughly investigated through the theory of homogenized materials and validated experimentally [22,23,24]. For soft materials for which the bulk modulus $K_{0}$ is much larger than G0=Re(G) (the real part of the shear modulus at ultrasound frequencies), only $K_{0}$ is significantly altered by the porosity. The sound speed *c* of longitudinal waves depends on the porosity ϕ via:(2)$$c\left(\phi \right)=c_{0}{\left(1+\frac{3K_{0}}{4G_{0}}\phi \right)}^{-1/2},$$
where $c_{0}$ is sound speed of longitudinal waves in the bulk, non porous material; c(ϕ) is thus an indicator of porosity and can be used as such to measure ϕ with some limitations we will prove below. To predict c(ϕ), we nevertheless require $K_{0}$ that is linked to $c_{0}$, $G_{0}$, and $\rho_{0}$ through: $K_{0}=c_{0}^{2}\rho_{0}-\frac{4}{3}G_{0}$ which reduces to $K_{0}\approx c_{0}^{2}\rho_{0}$ when $K_{0}\gg G_{0}$.

We measured $G_{0}$ using Dynamic Mechanical Analysis (DMA, not shown here): in the range of our application f=0.1−5 MHz, we obtained $G_{0}\approx \left(5\pm 3\right)$ MPa with a significant dissipation [Im(G)≈Re(G)]. We obtained the sound speed via the time of flight technique in bulk materials prepared under three different relative humidities (RH = 25, 50 and 75%). In principle, the solubilized water could affect the properties of the polymeric matrix, but we found no conclusive influence from such a maturing in different atmospheres, see Figure 6 top; we thus averaged the celerities to get $c_{0}=\left(1830\pm 90\right)$ m·s^−1^. The density was given above, Section 3.2.2.1: $\rho_{0}=\left(1.21\pm 0.03\right)\times 10^{3}$ kg·m^−3^.

We end up with $K_{0}=\left(4.0\pm 0.5\right)$ GPa $\gg G_{0}$ which indeed comforts the denomination of soft material. With $K_{0}/G_{0}∼O\left(10^{3}\right)$, Equation (Equation 2) represents a good description of the sound speed that is strongly altered by the presence of porosity, as evidenced by the solid red line of Figure 6 bottom, and actually strongly drops with the first signs of porosity until it reaches a pseudo-plateau in the range ϕ=0.2−0.5.

Figure 6 bottom shows the measurements of the sound speed at ultrasonic frequencies (performed in the frequency range f=0.5−5 MHz depending on the attenuation of the medium) as a function of porosity ϕ. Three sets of symbols are used: the open symbols refer to the volume fraction of calcite in the liquid formulation, whereas the red solid symbols represent the porosity estimated through the density measurements (Section 3.2.2.1, Figure 5); eventually, the black solid symbols use the porosity extracted from image analysis (Section 3.2.2.2, Figure 5).

At low porosity, the agreement between theory and experiments is poor, which results from the ill-explained density in this range of porosity, maybe due to the hygroscopic nature of the PEG-DA. In addition, the time-of-flight ultrasound measurement requires slightly squeezing the monolith between two transducers to ensure a good contact and may deform/collapse the porosity especially when the latter is low. We thus believe the sound speeds measured at low to moderate porosities are not reliable or reflect the uncertain nature of the sample. However, the agreement between theory and experiments is excellent at high ϕ whatever the way the porosity is expressed, with no fitting parameter. It demonstrates that the porous scaffold actually gathers all the ingredients to sustain very slow longitudinal acoustic waves (c≈150−200 m·s^−1^, see insert of Figure 6 bottom), namely some softness and compressibility, here provided by the high porosity of the matrix and a significant portion of voids.

## 4. Advanced Material Forming

The soft porous polymers we described are formulated initially as water-based suspensions with rheological properties that make them easily processable. Here, we demonstrate that we can indeed take advantage of such a processability in order to transform the liquid formulations into different categories of soft porous solids.

### 4.1. Porous Particles out of Emulsions

#### 4.1.1. Batch Emulsions

The water-based formulation can easily be dispersed into a fluorinated oil (FC-40) and upon addition of a surfactant (1%w in oil) and some mechanical energy—here strong manual stirring—an emulsion can be prepared. This emulsion is then exposed under UVs which turn the liquid droplets into solid particles; the dispersion is centrifuged to collect the supernatant which is dried as such or dried after acidic treatment which removes the solid calcite particles and opens the porosity.

In this procedure, we have not controlled precisely the amount of calcite nor the proportion of oil versus formulation but simply explored qualitatively the ability the produce spherical porous particles. The result is quite conclusive—see Figure 7—which shows SEM images of low and high porosity particles. Two main results emerge: the porosity is controllable via the amount of initially dispersed calcite, and the size polydispersity of particles is large, as expected, but could be improved to a large extent via advanced emulsification techniques [25].

#### 4.1.2. Microfluidic Engineered Emulsions

Another well established way to control the size distribution of particles in the range 10–100 μm, and possibly to enhance their structure and functionality, is to use microfluidic engineering in the droplet-based regime [26,27] in order to transform the liquid templates into advanced particles. Figure 8 (left) shows the geometry we used which forces the flow of the formulation to be squeezed in a neck and fragmented into droplets with the use of a non miscible oil (see bottom left picture of Figure 8). Here, we use FC-40 with surfactant and the PDMS chip was made fluorophilic in order to prevent the wetting of the formulation on the walls of the chip. It ensures a proper way to continuously generate droplets with no fouling of the microdevice. The latter also possesses a serpentine which permits us to increase the residency time of the droplets where they undergo UV-based curing in order to collect solidified particles at the outlet of the device. Typical flow diagrams have been established in order to map the flow behavior of the dispersion against the flow rates and satisfactory conditions are $Q_{d}<2$ μL·min^−1^ for the dispersed phase (the formulation) and $Q_{c}>50$ μL·min^−1^ for the continuous phase in order to generate particles that are spherical when popping out of the nozzle of the device. The method is indeed quite efficient: the right part of Figure 8 shows a SEM example of the well defined, monodispersed porous particles (size dispersity ≈5%) after acidic calcite dissolution and drying, along with the histogram of sizes before and after drying.

### 4.2. Soft Embossing

Here, we use a PDMS stamp that contains cylindrical cavities (diameter 200 μm, thickness up to 30 μm) laid on a square grid (or possibly any other pattern and grid) in order to confine the fluid formulation in between a substrate and the voids of the stamp, see Figure 9. Then, the formulation is exposed to UVs through the PDMS stamp, the stamp is removed, and solid patterns remain on the substrate. Eventually, an acidic treatment opens up the porosity. Two crucial steps make this approach feasible: first, the PDMS stamp is thoroughly degassed before contact in order to remove gas bubbles that could be trapped in the cavities [28]; then, the glass substrate is made acrylate-compatible by silane treatment in order to enhance the anchoring of the hydrogel on the substrate.

Figure 9 demonstrates that this approach is quite successful to create patterned substrates for which large scales (20 × 20 cm^2^) are easily achievable. Importantly, both the size of the porous cylinders and their repeating distance can be tuned independently; it is of course also possible to tune the lattice type, to generate disordered or amorphous structures, and ultimately to choose some structure factor to design a set of positions in order to investigate the role of correlation and order [29,30] on surfaces made of resonating sub-structures, namely the porous cylinders, and of course also to generate in-plane gradients of lattice parameter for acoustic metasurfaces [31].

## 5. Conclusions

We demonstrated here that the sacrificial route, whereby we formulate a mixture of PEG-DA oligomers with water and sacrificial particles, turn them into hydrogels through photo-polymerization, and remove the calcite particle via acidic dissolution, is interesting for controlling precisely the porosity we wish to engineer: the final porosity $\phi_{M}$ is equal to the volume fraction ϕ of sacrificial particles initially embedded in the formulation. This is especially true at high porosity (ϕ>0.1), whereas, at low porosity, we believe the very hygroscopic nature of the polymer leads to some water sequestration that may block the porosity. It could be interesting to investigate other PEG-DA oligomers with smaller molecular weight for which the water affinity decreases; however, the key point in our approach is to be able to remove the sacrificial particles and the by-products of dissolution and the permeability of polymeric scaffold is of prime importance for this process. There must be a trade-off of water affinity where it remains possible for the acid and salts to diffuse inside and outside the structure to remove the sacrificial particles along with a lesser affinity with water in order to also generate small porosities.

The second advantage of a water-based formulation, besides easy calcite removal, consists in its processability: the low viscosity and moderate yield-stress of the formulations make it fairly accessible for microfluidics or soft-embossing, which open up the route to advanced materials. We think here of acoustic metamaterials [31,32,33,34] for which this project was specifically designed, but a wealth of interesting possibilities exists: monoliths can be used as membranes or as biocompatible substrates for cell-growth or tissue engineering [35]; tailor-made porous particles are interesting for chromatography [5], chemical-delivery [27,36], of inclusions for tuning the mechanical properties of a matrix; micro-patterns can be included into microfluidic chips for enhanced functionalities such as filtering, dialysis, etc. [37,38].

## Figures and Tables

**Figure 1 polymers-12-01008-f001:**
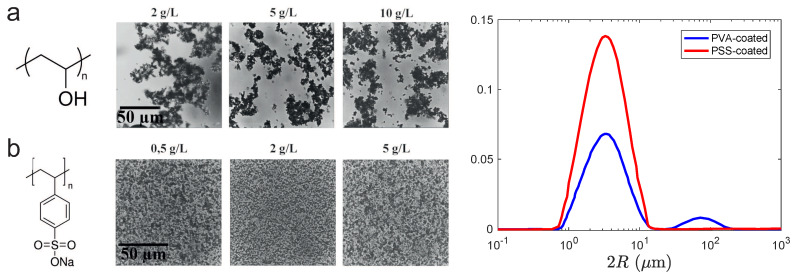
**Left**: (**a**) chemical formula of poly(vinyl alcohol) (PVA) and pictures of formulations containing calcite and PVA (concentration given above the image); (**b**) chemical formula of poly(sodium 4-styrenesulfonate) (PSS) and pictures of formulations containing calcite and PSS. In both cases, the mass fraction of calcite is 2%. **Right**: size distribution of CaCO_3_ powders in water with PSS or PVA (2% w/w of particles in a 2 g·L^−1^ PVA or PSS solution).

**Figure 2 polymers-12-01008-f002:**
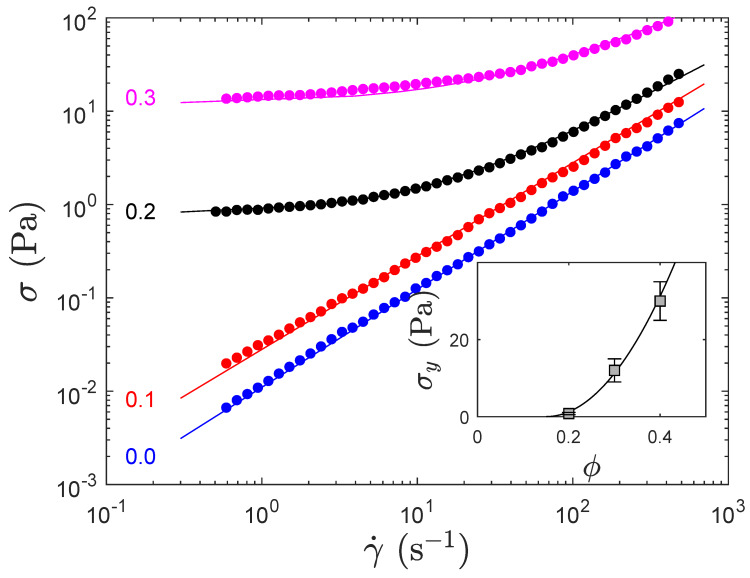
Flow curves of different PSS-stabilized formulations where the volume fraction of calcite is noticed in the figure. Some yield-stress develops between 0.1 and 0.2 calcite volume fraction. The straight lines describe a purely Newtonian fluid whereas above 0.2 volume fraction of calcite, the fluid obeys a Herschel–Buckley behavior ($\sigma =\sigma_{y}+K{\dot{\gamma }}^{\alpha }$). Insert: Yield stress $\sigma_{y}$ that develops above 0.1–0.2 volume fraction of calcite. The solid line is a quadratic fit (a guide) that suggests that the critical yield stress occurs at ≈0.15.

**Figure 3 polymers-12-01008-f003:**
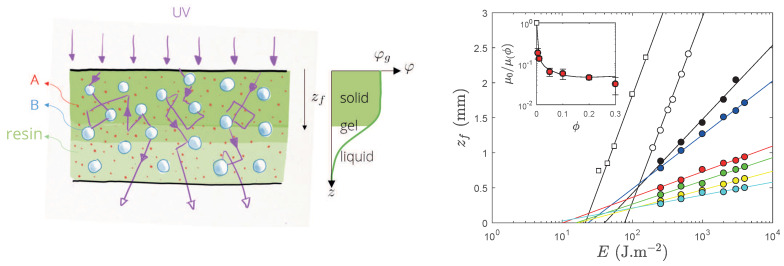
**Left**: sketch of the FPP geometry where a thick slab containing a formulation [resin + photo-initiator (A) + particle (B)] is exposed to a collimated UV-light and thickness-dependent profile of photo-conversion of the resin develops in depth (inspired by Ref. [11], where φ denotes the degree of monomer conversion, and $\varphi_{g}$ the gel point that occurs upon a given cross-link value). **Right**: Cured thickness $z_{f}$ during frontal polymerization of the PEG-DA formulations against UV dose for several cases with no calcite (open symbols, circle for no degassing, square for degassed polymer) or with different volume fractions ϕ of calcite but no degassing ($\phi =5\times 10^{-3},\;1\times 10^{-2},\;5\times 10^{-2},\;0.1,\;0.2,\;0.3$, black, blue, red, green, yellow, cyan symbols respectively). Insert: slope of the front propagation curve $\mu {\left(\phi \right)}^{-1}$ normalized by the one of the bare resin; the line is a model not discussed here.

**Figure 4 polymers-12-01008-f004:**
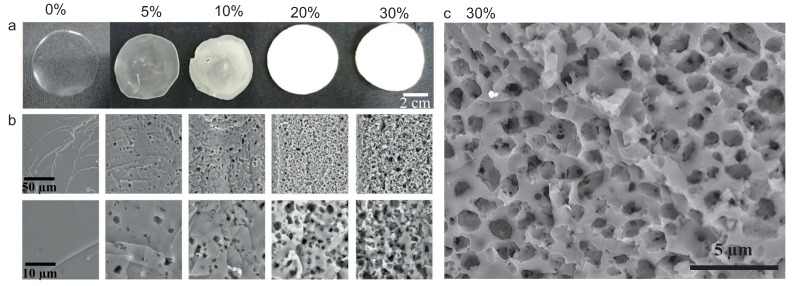
(**a**) monoliths prepared with volume fractions of calcite between 0 and 0.3, observed here after photo-polymerization, calcite removal by acidic dissolution, and drying; (**b**) SEM views of cut samples of the same monoliths for two magnifications; (**c**) close up of a porous polymer with 0.3 porosity.

**Figure 5 polymers-12-01008-f005:**
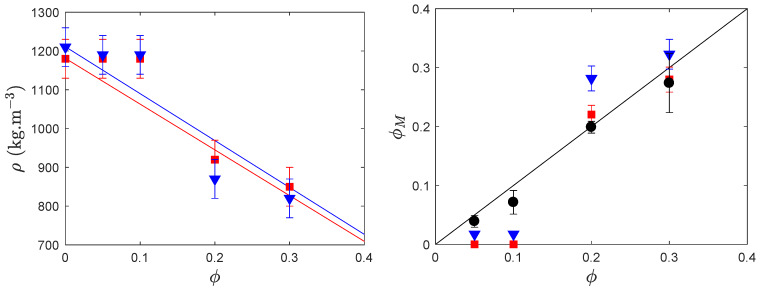
Left: Calculated densities of porous samples dried under ambient conditions (red symbols) or in an oven (blue symbols) as a function of the initial volume fraction of CaCO${}_{3}$. The lines are calculated using only the density $\rho_{0}$ of the bulk polymer: $\rho =\rho_{0}\left(1-\phi \right)$. Right: Porosity $\Phi_{M}$ as a function of initial volume fraction ϕ of CaCO_3_ obtained from image analysis (Section 3.2.2.2, black symbols) or density measurements (Section 3.2.2.1, blue and red symbols for oven or air-dried samples); the black line represents the expected porosity assuming an isotropic drying.

**Figure 6 polymers-12-01008-f006:**
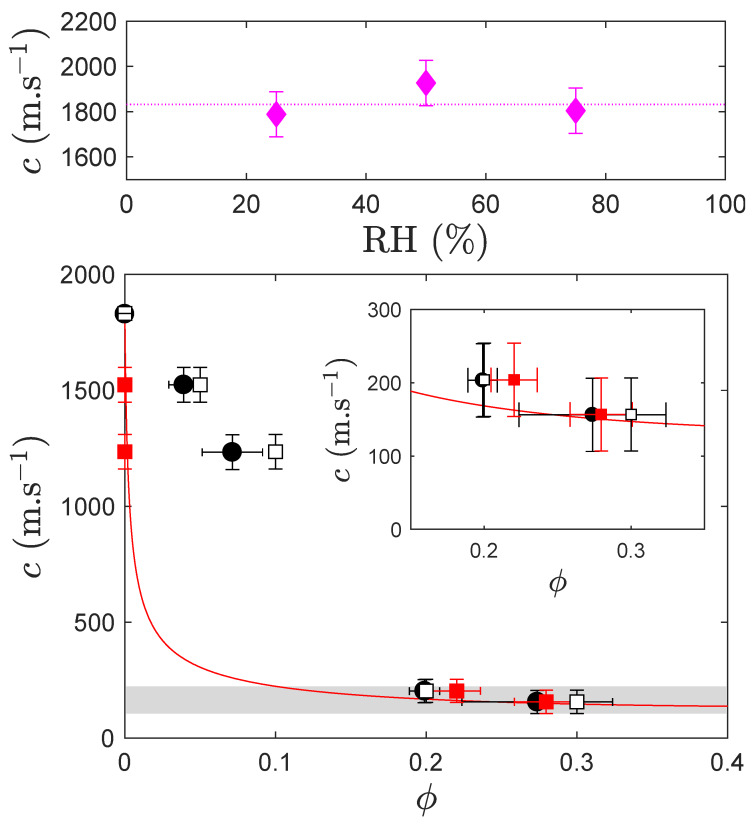
Top: sound speed of a bare polymer (ϕ=0) as a function of relative humidity (RH). Main: sound speed a function of the porosity in the polymer. The open symbols refer to as the nominal volume fraction of calcite in the initial formulation; the black solid symbols show the porosity estimated from image analysis; the red solid symbols show the porosity estimated from density measurements. The gray region highlights the high porosity zone where all measurements match. The solid line is derived from the theory of effective media given by Equation (Equation 2) with no fitting parameter. Insert: close-up at high porosity.

**Figure 7 polymers-12-01008-f007:**
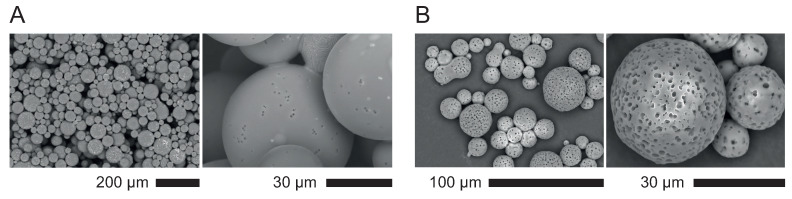
Batch particles produced by manual emulsification followed by UV-curing. (**A**) low density of calcite (before and after removal of calcite, left and right respectively where the calcite particles show as bright spots); (**B**) high density of calcite after its removal.

**Figure 8 polymers-12-01008-f008:**
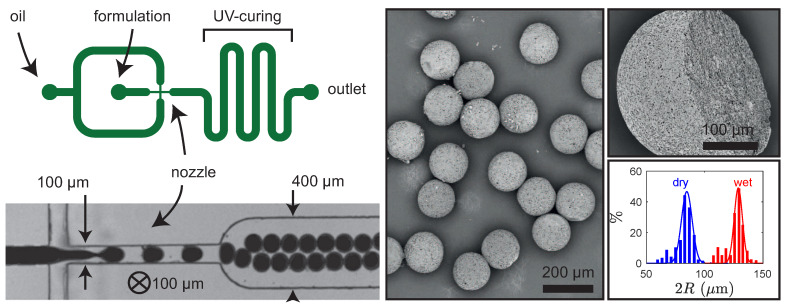
**Left**: schematic principle view of the microfluidic setup used to produced particles. **Right**: SEM views of the particles after calcite dissolution and drying, and histogram of size reduction after drying.

**Figure 9 polymers-12-01008-f009:**
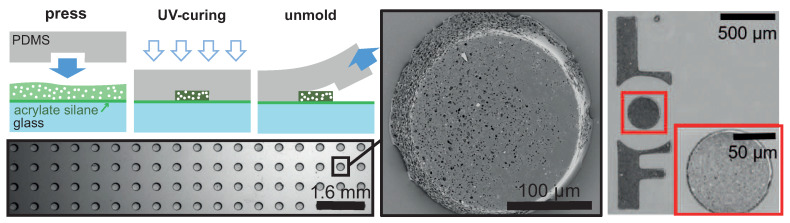
Sketch of the soft embossing process with a PDMS stamp designed with micro-cavities, here cylindrical cavities on a square lattice. The stamp squeezes the formulation in the cavities and may lead to well defined patterns on a substrate after UV-curing and acidic treatment. The pictures show a small part of the patterned substrate and a magnification on one cylinder, and also the logo of our lab (LOF) soft-embossed the same way.

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
