# Peer review of "A Sacrificial Route for Soft Porous Polymers Synthesized via Frontal Photo-Polymerization"

_polymers, 2020, doi:10.3390/polym12051008_

Round 1

Reviewer 1 Report

In the manuscript “A Sacrificial Route for Soft Porous Polymers Synthesized via Frontal Photo-Polymerization”, Turani-i-Belloto et al. studied the effect of adding sacrificial CaCO3 particles in the frontal photo-polymerization. Both polymerization kinetic and resulting porosity as a function of CaCO3 concentration are examined. The advantage of aqueous solution is also demonstrated by conducting the photo-polymerization in batches, in micro-devices, as well as on surfaces. The paper is quite informative and easy to follow. I recommend acceptance after a few minor comments have been addressed:

  1. The language needs to be further polished throughout the paper. For example, there are sentences mixed with “the best”, “the more” and “the most” (line 94-95). The use of present tenses and past tenses is not consistent in the paper. There are two “Eventually” in section 3.1. There are also misleading typos such as “resist” in Figure 3.
  2. In Figure 1, is the legend mislabeled? Why can one single plot (blue) demonstrate two data sets: native or PVA?
  3. How the cured thickness, zf, is measured? How is the frontier of fully-conversed solid hydrogel defined in measurements?
  4. In line 125, authors claimed to have checked that only photo-initiator absorbs UV radiation. Is there previous study supporting that water, PSS, or PEG-DA absorb do not absorb UV radiation?
  5. Please add reference where the value “2.5 mm” comes from, in line 127.
  6. In Figure 9, why does the inset zoom-in picture of LOF seems very smooth and not porous? The picture in the middle appears to be more porous under the similar scale.

Author Response

In the manuscript “A Sacrificial Route for Soft Porous Polymers Synthesized via Frontal Photo-Polymerization”, Turani-i-Belloto et al. studied the effect of adding sacrificial CaCO3 particles in the frontal photo-polymerization. Both polymerization kinetic and resulting porosity as a function of CaCO3 concentration are examined. The advantage of aqueous solution is also demonstrated by conducting the photo-polymerization in batches, in micro-devices, as well as on surfaces. The paper is quite informative and easy to follow. I recommend acceptance after a few minor comments have been addressed:

We thank the Referee for the positive critique and will give below a detailled answer to her/his remarks.

The language needs to be further polished throughout the paper. For example, there are sentences mixed with “the best”, “the more” and “the most” (line 94-95). The use of present tenses and past tenses is not consistent in the paper. There are two “Eventually” in section 3.1. There are also misleading typos such as “resist” in Figure 3.

We fixed the repetitions, the lisleading ‘resist’, and we also tried our best to polish the English but remember we remain French native speakers! Besides, I must admit than I do not like thorough homogenization. I believe the tenses need not be homogeneous along the paper (even through the have to concord); I even think it makes the reading more dynamic.

In Figure 1, is the legend mislabeled? Why can one single plot (blue) demonstrate two data sets: native or PVA?

No, there is actually no difference between the data from the native dispersion and the data of PVA coated particles. But you are right, it is misleading and I modified the legend in the figure and explained in the text this point.

How the cured thickness, zf, is measured? How is the frontier of fully-conversed solid hydrogel defined in measurements?

The cured thickness is measured sidewise from a cross-section of a cut sample using stereo-microscopy or optical microscopy depending on the size of the sample. We added this point in the text.

In line 125, authors claimed to have checked that only photo-initiator absorbs UV radiation. Is there previous study supporting that water, PSS, or PEG-DA absorb do not absorb UV radiation?

We actually measured systematically the absordance of all species (except microparticles of course) and found that at 365 nm, only the photo-initiator absorbs light (basically, for the latter, we measured the beer lambert law) and this is how we measured the value  value. We made this point more clear in the text.

Please add reference where the value “2.5 mm” comes from, in line 127.

See above

In Figure 9, why does the inset zoom-in picture of LOF seems very smooth and not porous? The picture in the middle appears to be more porous under the similar scale.

It is probably due to the fact that the picture in the middle was acquired using Scanning Electron Microscpy whereas the insert was acquired with large-scale (low resolution) optical microscopy. Yet, we can neverthless notice some granulometry that reveals the porosity.

Reviewer 2 Report

The submitted manuscript describes the preparation of porous materials. Although is topic is well known the authors present a comprehensive study of this particular system. The work is well done and might be published as it is.

Author Response

Many thanks